# Properties of a Novel *Salmonella* Phage L66 and Its Application Based on Electrochemical Sensor-Combined AuNPs to Detect *Salmonella*

**DOI:** 10.3390/foods11182836

**Published:** 2022-09-13

**Authors:** Changbin Li, Yuanshang Wang, Jia Wang, Xiaohong Wang

**Affiliations:** 1College of Food Science and Technology, Huazhong Agricultural University, Wuhan 430070, China; 2College of Food and Biology Engineering, Henan University of Animal Husbandry and Economy, Zhengzhou 450046, China; 3Key Laboratory of Environment Correlative Dietology, Huazhong Agricultural University, Wuhan 430070, China

**Keywords:** phage, *Salmonella* Typhimurium, electrochemical sensor, gold disc electrode, biological properties

## Abstract

*Salmonella* is widespread in nature and poses a significant threat to human health and safety. Phage is considered as a new tool for the control of food-borne pathogens. In this study, *Salmonella* phage L66 (phage L66) was isolated from sewage by using *Salmonella* Typhimurium ATCC 14028 as the host bacterium, and its basic properties were obtained by biological and bioinformatics analysis. Phage L66 had a broad host spectrum, with an optimal infection complex of 0.1 and an optimal adsorption rate of 90.06%. It also exhibited thermal stability between 30 °C~60 °C and pH stability pH from 3 to 12, and the average lysis amount was 46 PFU/cell. The genome sequence analysis showed that the genome length of phage L66 was 157,675 bp and the average GC content was 46.13%. It was predicted to contain 209 genes, 97 of which were annotated with known functions based on the evolutionary analysis, and phage L66 was attributed to the *Kuttervirus* genus. Subsequently, an electrochemical sensor using phage L66 as a recognition factor was developed and the working electrode GDE-AuNPs-MPA-Phage L66 was prepared by layer-by-layer assembly for the detection of *Salmonella*. The slope of the impedance was 0.9985 within the scope from 20 to 2 × 10^7^ CFU/mL of bacterial concentration. The minimum detection limit of the method was 13 CFU/mL, and the average spiked recovery rate was 102.3% with a relative standard deviation of 5.16%. The specificity and stability of this sensor were excellent, and it can be applied for the rapid detection of *Salmonella* in various foods. It provides a phage-based electrochemical biosensor for the detection of pathogenic bacteria.

## 1. Introduction

*Salmonella* is second only to *Campylobacter* as a zoonotic pathogen, of which *Salmonella* Typhimurium and *Salmonella enterica* are the worst [1,2]. *Salmonella* belongs to the Gram-negative bacteria, which have flagella for activity commonly [3]. *Salmonella* can cause infections such as diarrhea, fever, enteritis, and other extraintestinal complications [4]. With the overuse of antibiotics, more and more strains of *Salmonella* resistant bacteria have emerged, posing challenges to clinical medicine and food safety [5,6]. Therefore, it was significant to find antibiotic alternatives and establish methods for *Salmonella* detection.

Bacteriophage (also phage) is a virus that consists of a protein shell and DNA. It binds to the receptor of its host and is characterized by high specificity. Phages are easy to obtain and are highly resistant to temperature, pH, and ionic strength [7,8]. Recently, phages have been used as excellent bio-recognition probes because of their ability to distinguish between living and dead cells, and have shown promise for developing detection methods for bacteria [9,10].

In the field of bacterial detection, the biosensor is defined as an analytical device consisting of biological recognition elements and signal converters, which enable quantitative detection of targeted components converted into detectable signals such as light and electricity [11].

There are electrochemical biosensors [12,13], piezoelectric biosensors [14], surface plasmon resonance sensors [15,16], and photosensitive sensors [17,18], etc. Among these sensors, electrochemical biosensors, especially phage-based biosensors, are now used because of their simplicity of operation, high selectivity, fast detection speed, and easy miniaturization [19,20,21,22].

In this study, a novelty phage was screened to develop a detection method for *Salmonella*. The function of phage was investigated, and it was combined with functionalized gold nanoparticles (AuNPs), and utilized on a gold disc Electrode (GDE) to build a biosensor. This study provides an opportunity and theoretical basis for the detection method of *Salmonella* using phage as a new biological recognition element.

## 2. Materials and Methods

### 2.1. Chemical and Materials

Potassium ferricyanide (K_3_[Fe(CN)_6_]), potassium ferrocyanide (K_4_[Fe(CN)_6_]), hydrochloric acid, phosphate buffer saline (PBS), and potassium chloride were obtained from Bioshop (Burlington, ON, Canada). Luria−Bertani (LB) Broth and Agar were obtained from Haibo Biotechnology (Qingdao, China). N-hydroxy sulfosuccinimide (NHS) and 1-ethyl-3-[3-dimethylaminopropyl] carbodiimide hydrochloride (EDC) were obtained from Sigma-Aldrich (Oakville, ON, Canada). Bovine serum albumin (BSA), chloroauric acid tetrahydrate (HAuCl_4_·4H_2_O), sodium citrate, and 3-mercaptopropionic acid (MPA) was obtained from Aladdin (Shanghai, China). Alumina (0.3 and 0.05 μm), polishing cloths, gold disc electrodes (GDE, 3 mm diameter), glycerol electrode, and platinum wire electrodes were obtained from Chenhua (Shanghai, China). All chemical reagents were analytically pure. The water used in the experiment was deionized distilled water with a resistivity of 18.2 MΩ·cm.

### 2.2. Bacterial Strains, Bacteriophage, and Growth Conditions

Bacteria strains were stored in 20% glycerol and frozen at −80 °C. Appendix A supported detailed information. *Salmonella* Typhimurium (*S.* Typhimurium) ATCC 14028 and others were incubated in LB broth (containing 0.7% Agar) at 37 °C for 18 h. Several uniform and consistent colonies were selected and then incubated for 8–12 h at 37 °C in LB broth with shaking [23].

Using *S.* Typhimurium 14028 as the host strain, phage L66 was first isolated from local sewage in Zhengzhou, Henan province, China. After purification five times, the phage titer was determined using the double-layer agar plate method [24,25]. Purified phage was stored in 20% glycerol at −80 °C.

### 2.3. Characteristics of Phage L66

#### 2.3.1. Morphological Analysis

The phosphotungstic acid negative staining method was employed to observe the morphology of phage L66 [26]. Briefly, phage L66 suspension was ultracentrifuged at 30,000 rpm for 1 h and suspended in SM buffer solution. The phage L66 was stained with phosphotungstic acid solution with pH 7 for 10 min and dried at 25 °C. The morphology of the phage L66 was imaged by transmission electron microscopy (TEM) at 75 kV.

#### 2.3.2. Host Lytic Range

The host lytic range of phage L66 was evaluated by the spot test method, and the bacteria strains used were shown in Appendix A. In brief, initially, the bottom layer was created with LB broth containing 1.5% agar, the overlay mixed with bacterial suspension and 3.8 mL LB broth at 42 °C. The phage L66 (5 μL, 10^6^ PFU/mL) was spotted on the tested plates, then incubated at 37 °C for 18 h. Finally, the lysis ability of phage was evaluated based on phage spots [27]. 

#### 2.3.3. Optimal Multiplicity of Infection

The multiplicity of infection (MOI) is defined as the potency ratio of phages with host bacteria. Based on a certain MOI (0.001, 0.01, 0.1, 1, 10, 100), a 500 μL phage was added with an equal amount of host bacteria, then incubated in LB with constant shaking at 37 °C for 210 min, and centrifugated at 8000 rpm at 4 °C for 10 min. The titer of supernatant was determined to follow the reported method [28]. The highest titer was recognized as the MOI of phage L66.

#### 2.3.4. Adsorption Rate

According to the measured optimal MOI, 5 mL of phage was mixed with 5 mL of host bacteria. The mixture was then co-cultivated with 160 rpm at 37 °C. Starting from 0 to 45 min, 300 μL of the mixture was taken every 5 min and centrifuged at 8000 rpm for 60 s with 4 °C, then, the phage titer was determined and the adsorption rate was calculated.

#### 2.3.5. One-Step Growth Curve

One-step growth curve can display the incubation period, burst period, and burst size of bacteriophage [29]. Based on the optimal MOI, 500 μL of phage was mixed with the same volume of host bacteria. Then, the mixture was incubated with constant shaking at 37 °C for 25 min, followed by centrifuging at 12,000 rpm for 2 min. The mixture was washed three times with LB and added to 10 mL of LB, and then incubated at 37 °C for 3 h. Subsequently, 300 μL incubation solution was taken out every 10 min and centrifuged (13,000 rpm, 4 °C) for 1 min. The phage titer was determined using the previously mentioned method. The latent period refers to the time from the first burst to adsorption. The burst size defined the ratio of the final phage titer with the initial.

#### 2.3.6. pH and Temperature Stability

Phage L66 was co-cultured at various pH and temperatures to evaluate its tolerance against environmental stresses. To evaluate the pH stability, phage (100 μL, 10^7^ PFU/mL) was added into 900 μL LB pre-adjusted from pH 2 to 13, which was incubated for 2 h at 37 °C. The mixture (100 uL) was mixed with 100 uL host strain the titer of phage was determined by double-layer plate method. For thermal stability, the mixture was co-cultured at 37 °C for 2 h, and then diluted to 10^7^ PFU/mL with PBS buffer at a preset temperature, ranging from 30 to 80 °C for 30 min and 60 min.

### 2.4. Genome Sequencing, Annotation, and Comparison Analysis of Phage L66

The genomic DNA of phage L66 was extracted and purified using the method of Protease K/SDS and phenol-chloroform [30]. The genome-wide association was sequenced on the HiSeq platform and assembled using MicrobeTrakr plus. The BLASTn of NCBI was applied for sequence similarity alignment, and RAST was utilized for functional annotation.

The genome map of phage L66 was obtained using the BLAST Ring Image Generator (BRIG), and the phylogenetic tree was constructed and displayed using MEGA 7, based on the sequence of terminal large subunit enzymes [31,32]. Antibiotic Resistance Genes Database (ARDB) and Virulence Factor Database (VFDB) were used to identify the antibiotic resistance and virulence factor genes.

### 2.5. Construction of Phage L66-Based Electrochemical Sensors

#### 2.5.1. Activation of Electrode and Preparation of Gold Nanoparticles

To obtain a uniform surface before modifying the electrode, the GDE was first physically polished by alumina-water pastes with particle sizes of 0.3 and 0.05 μm on a polishing cloth separately, followed by washing thoroughly with deionized water, ethanol, and acetone by ultrasound for one minute, respectively [33,34]. Then, the electrode was electrochemically etched by Cyclic voltammetry (CV) between −0.2 and 1.0 V with a scan rate of 100 mV/s for 100 cycles in a 0.5 M H_2_SO_4_ solution. The experiment was conducted under a nitrogen atmosphere. Subsequently, the electrodes were assayed with Cyclic voltammetry and Electrochemical impedance spectroscopy (EIS) determinations in a mixture of potassium ferricyanide and potassium ferricyanide at each step. The process was measured in a three-electrodes system with GDE (diameter was 3 mm) being the working electrode, Ag/AgCl (3 mol/L KCl) as the reference electrode, and platinum wire acting as the counter electrode [35,36,37].

Gold nanoparticles with a diameter of nm were prepared by sodium citrate reduction using chloroauric acid as raw material [38]. The morphology and size distribution of the gold nanoparticles were characterized by scanning electron microscope (SEM), UV-Vis spectrophotometer, and Zetasizer nano instrument, respectively [39,40].

#### 2.5.2. Preparation of GDE-AuNPs-MPA-Phage L66 Composite Electrode

In Section 2.5.1, we processed the GDE and obtained the gold nanoparticles, then the GDE-AuNPs-MPA-Phage L66 composite electrode was prepared through layer-by-layer assembly. Initially, a layer of gold nanoparticles was deposited on the gold electrode by one-step constant potential. Subsequently, a carboxyl group was formed on the surface of the gold electrode using 3-mercaptopropionic acid through the bonding mechanism of the Au-S bond. Then, the amino group of EDC-NHS was used to condense with the carboxyl group, because the reaction produced a lively and unstable amide ester intermediate, which can be easily replaced by amine on the surface of the phage [41,42,43].

For a covalently bound phage to a gold electrode, 0.01 mol/L NHS and 0.05 mol/L EDC were added to 1 mL PBS solution, and then phage solutions with different concentrations from 2.5 × 10^3^ PFU/mL to 2.5 × 10^10^ PFU/mL were added, respectively. Then, phage L66 were immobilized on the surface of GDE-AuNPs-MPA due to the amide bonding [44]. The electrodes were rinsed completely with sterile deionized water after measuring different concentrations of phage.

In all reaction systems, a mixture of potassium ferricyanide and potassium ferricyanide with potassium chloride as solvent was used as a redox substrate for electrode determination. The concentrations of solutions were designed to be 2.5 and 5 mmol/L, and the concentrations of potassium chloride were 0.05, 0.1, and 0.2 mol/L, respectively [45]. In the end, potassium chloride solution with a concentration of 0.2 mol/L containing 5 mmol/L potassium ferricyanide and ferricyanide (1:1) was used for the measured impedance values.

In the whole process of the construction of a biosensor, parameters used were optimized, such as working voltage range of cyclic voltammetry, electrodeposition time, EDC-NHS activation time, the concentration of phage, time of phage immobilization, and block time of BSA, respectively.

#### 2.5.3. The Specificity and Stability Evaluation of the Sensor

To verify the specificity of the sensors, *Listeria monocytogenes*, *Staphylococcus aureus*, *Escherichia coli*, *Bacillus subtilis*, and different serotypes of *Salmonella* with the same concentration of 2.5 × 10^6^ CFU/mL were used for evaluation. The stability of biosensors were evaluated by measuring the impedance at different pH from 3 to 11, and the temperature conditions were also measured at 4 °C and 25 °C every three days for a total of 21 days [46].

### 2.6. Determination of Salmonella in Contaminated Food

To test the applicability of the sensor, detection of *Salmonella* in spiked milk, eggs, and chicken was assessed, respectively. Food samples were purchased from a local supermarket and stored in a 4 °C refrigerator. Before use, spiked milk was mixed directly, eggs were shelled and mixed, the chicken was made into a homogenate, and all samples were irradiated under a UV lamp (15 W) for 30 minutes to ensure sterility. Then milk, eggs, and chicken were spiked with *Salmonella* at final concentrations of 20, 2.0 × 10^2^, and 2.0 × 10^3^ CFU/mL, respectively, whereas un-spiked samples were tested as negative control [47].

### 2.7. Statistical Analysis

All experiments were completed three times independently in parallel. The data measured were processed with analysis of variance with SPSS version 25.0 (Armonk, NY, USA). The recovery range and relative standard deviation (RSD) were obtained to evaluate precision, meanwhile, error bars were shown on the graph.

## 3. Results and Discussion

### 3.1. Characterization of Phage L66

#### 3.1.1. Morphology of Phage L66

In this study, a new phage L66 was screened with *S.* Typhimurium 14028 as the host bacterium according to methods Section 2.2 Scanning electron microscopy (SEM) micrographs showed that the phage had a symmetry icosahedral head (diameter 83.0 ± 2.0 nm) and a tail (length 113.0 ± 2.0 nm) (Figure 1A). This indicated that phage L66 was consistent with the microscopic morphology of *Myoviridae* and had a typic structure of *Caudovirales* [48].

#### 3.1.2. Host Range Properties of Phage L66

The ability of phage L66 to lyse 49 strains (including 34 strains of *Salmonella*) of different bacteria was evaluated, and the results showed that it could lyse 31 strains of *Salmonella*, accounting for 91.2%, but none had lytic capability similar to other genera such as *S. aureus*, *Vibrio parahaemolyticus,* and *Listeria monocytogenes* (Appendix A). Phage L66 was assessed as a broad-spectrum potent phage, which has the potential to be used for the prevention and control of *Salmonella*.

#### 3.1.3. Basic Biological Characteristics of Phage L66

To further examine the inhibitory activity of the phage, different titers of phage L66 were incubated with host *S.* Typhimurium 14028 at 37 °C for 3.5 h. Figure 1B showed that the optimal multiplicity of infection of the phage L66 was 0.1. As indicated in Figure 1C, the adsorption of phage L66 on *S.* Typhimurium ATCC 14028 displayed a significant increasing trend within 20 min, and peaked reached 90.06% at 20 min. After that, the adsorption decreases rapidly. Generally, the short adsorption time and high adsorption rate indicate that phage L66 had vigorous activity. The kinetics process of the capture, replication, and release of *S.* Typhimurium ATCC 14028 by phage L66 were shown in Figure 1D. The latent period of phage L66 was about 20 min, which was shorter than that of reported relevant *Salmonella* phages [49,50]. Phage with a short latent is suitable for biological control because it can lyse more bacteria. Subsequently, the amount of phage increased dramatically in the subsequent 110 min, followed by stable growth up to 140 min. The average burst size of phage L66 was approximately 46.13 PFU/CFU.

The pH and thermal tolerance of phage L66 were evaluated as essential application indicators. The activity of phage L66 sharply decreased when pH < 3.0 or pH > 12.0, but it was highly stable over a wide range of pH from 3.0 to 12.0 with the titer greater than 7 log PFU/mL (Figure 1E). The phage potency was essentially unchanged between 30 °C and 50 °C. The phage potency was reduced by about 21.3% when it was incubated at 60 °C for 60 min. When the temperature was over 70 °C, the titer decreased by approximately 40% in 30 min, and the phage inactivated when the temperature was at 80 °C for 60 min (Figure 1F). Compared with published reports, phage L66 had similar pH and thermal stability [51,52].

### 3.2. Bioinformatics Analysis of Phage L66 Genome

#### 3.2.1. Analysis of the Phage L66 Genome

To provide a genetic background for the application of phage L66 in food safety control, the genome of phage L66 was analyzed. The results showed that phage L66 had a double-stranded DNA genome of 157,675 bp and GC content of 44.69%, which was consistent with the feature of *Myoviridae*. In total, 209 open reading frames (ORFs) were predicted, and of them, 97 could be assigned a putative function. The ORFs were allocated into four functional categories such as DNA packaging and nucleic acid metabolism, structural proteins, proteins related to cleavage, and tail-related proteins [53]. We could not find any genes related to lysogenicity. The results of phage L66 prediction and annotation were shown in Figure 2A.

#### 3.2.2. Evolutionary Analysis of Phage L66

Based on The International Committee on Taxonomy of Viruses (ICTV) classification, 18 phages with the highest sequence score of phage L66 gene were screened from the NCBI database [54]. The evolutionary relationships were analyzed based on the attribution of these phages. Phylogenetic analysis of the terminase large subunits revealed that phage L66 had a close evolutionary relationship with *Salmonella* phage SenASZ3 and *Salmonella* phage SE14 (Figure 2B). Combined with the electron microscopic morphology of phage L66, it was determined that phage L66 belonged to the order *Caudovirales*, family *Ackermannviridae*, subfamily *Cvivirinae*, a branch of the genus *Kuttervirus* (ICTV 2021) [55]. Website comparison showed that the virulence factor and antibiotic resistance gene were not found in the genome of phage L66, which confirmed the safety of phage for therapeutic and pathogenic bacteria control from a genetic background.

### 3.3. Construction and Evaluation of Phage L66-Based Sensors

According to the biological properties and genetic analysis of phage L66, an electrochemical biosensor was assembled based on phage L66 and exhibits great potential in the detection of *Salmonella*. In this study, a GDE-AuNPs-MPA-END/NHS-phage biosensor was constructed using layer-by-layer assembly with GDE as the substrate. The entire fabrication process was illustrated in Figure 1.

The GDE was responsible for providing the base platform. Firstly, gold nanoparticles were deposited on GDE by one-step constant potential, which mainly improves the efficiency of electron transfer; then, MPA was used to form exposed carboxyl groups, followed by EDC-NHS, which provided a bridge for the linkage of phage by covalent binding [56]. The phage was then bound to the exposed carboxyl group through the amino group, and the carboxyl group not bound was blocked with BSA. Finally, *Salmonella* was captured according to the specific interaction between phage and host bacterium [57]. The performance of biosensors can be obtained by optimization of construction parameters. Subsequently, specificity and stability of the biosensor were evaluated, and application for detection of *Salmonella* in spiked samples was studied.

#### 3.3.1. Optimization of Fabrication Conditions of Phage L66-Based Biosensor

**Size and distribution of gold nanoparticles**. The size and homogeneity of the gold nanoparticles had a significant impact on the performance of the sensor. Scanning electron micrographs (SEM) of the prepared gold nanoparticle were spherical with uniform distribution (Figure 3A). Figure 3B showed that the maximum UV absorption peak of the gold nanoparticles prepared in the laboratory is 520 nm, which was consistent with the characteristic peak of nanogold solution [58]. Figure 3C showed that the gold nanoparticles had a uniform size of 18 nm. In Figure 3D, the potential of gold nanoparticles was −29.7 mv, which was consistent with the properties of the solution.

**Operating voltage range.** The range of operating voltage is an important parameter of the CV method. CV scans were performed using bare electrodes in the electrolyte, and different potential ranges of 0 to 0.5 V, −0.1 to 0.6 V, −0.1 to 0.7 V, −0.1 to 0.8 V, −0.2 to 0.6 V, and −0.2 to 0.7 V were set for characterization using the CV method. The potential range of −0.2 to 0.6 V was chosen based on the symmetry and regularity of the graph.

**Deposition time of gold nanoparticles.** The loading of AuNPs on the electrode was controlled by adjusting the electrodeposition time for 5 min, 10 min, 20 min, and 30 min, respectively. The current peak increased with the deposition time in 40 min, the impedance achieved a minimum at 40 min, and the corresponding gold nanoparticles coverage reached a maximum. Finally, the deposition time was selected as 40 min (Figure 4A).

**The incubation time of 3-Mercaptopropionic.** 3-Mercaptopropionic acid and EDC/NHS perform a pivotal role in immobilization numbers and level of phage, and their incubation time directly determines the sensor’s performance. As shown in Figure 4B, increasing amounts of 3-mercaptopropionic acid were immobilized 40 min before, reaching maximum loading at 40 min and then essentially stability; therefore, 40 min was chosen.

**The incubation time of EDC/NHS.** Figure 4C explains the effect of EDC/NHS incubation time on the biosensor. The activation times were set to 10 min, 20 min, 30 min, 40 min, 50 min, and 60 min. With the increase in the activation time, the current signal gradually increased until the current decreased after 30 min, likely due to the number of activated groups reaching saturation at 30 min, after which the electrode had a slight signal decrease. Therefore, 30 min was chosen as the optimal activation time.

**Concentration and incubation time of phage L66.** The concentration and incubation time of the phage solution affects the phage fixation and the detection limit of the sensor. The effect of phage concentrations from 2.5 × 10^3^ to 2.5 × 10^8^ PFU/mL on the current signal was investigated. As shown in Figure 4D,E, the larger the phage concentration, the more phage bound accompanying the current signal decreased, and the change was relatively small when the concentration changed from 2.5 × 10^6^ to 2.5 × 10^8^ PFU/mL. Considering the sensitivity of the sensor, phage concentration was selected as 2.5 × 10^6^ PFU/mL. The current value was unchanged after 60 min of fixation time. Therefore, 60 min was selected. This was mainly due to the fact that the phage was bonded on an electrode with its head and the tails, which were used to capture the host bacterium, so that no more naked sites were employed when the phage combined the carboxyl group in the maximum amount [59].

**Blocking time of BSA.** After phage immobilization on the electrode, unadsorbed sites needed to be blocked with BSA protein. Under the same conditions, the change of resistance was measured every 10 min from 0 min to 90 min, respectively. As shown in Figure 4F, the signal decreased with the increase of closure time and reached a plateau at 60 min, after which it remained unchanged. Therefore, the incubation time of BSA was chosen as 60 min.

#### 3.3.2. Characterization of Phage L66-Based Biosensor

The working electrodes were prepared according to previous experimental conditions and characterized electrochemically step by step. The CV and EIS were performed to verify the assembly process of the biosensor preparation. As shown in Figure 5A, curve a represents the bare electrode. When the AuNPs were combined with the electrode, the signal increased significantly, and the resistance decreased. This was mainly due to the excellent conductivity of the AuNPs, which can accelerate the electron transfer on the electrode interface [60].

After 3-mercaptopropionic acid was bound to the electrode, the current signal was reduced and the resistance increased. We attributed this situation to the negative charge of the carboxyl group, which repelled the redox probe with the same negative charge and hindered the electron transfer [61]. The curve reflects the situation after activation by EDC-NHS. The carboxyl group on the electrode surface is replaced by the ester group, which accelerates the electron transfer of the redox probe, and makes the current signal increase significantly. Curves e and f of Figure 5A indicate the changes of potential and resistance after electrode binding phage and BSA protein, respectively. Phage and protein are non-conductive biomolecules, which mask the electron transfer site and hinder the electron transfer; as a result, the currents were significantly reduced and the impedances increased subsequently [62].

Figure 5B showed the electrochemical impedance characterization of the biosensor, represented as a Nyquist graph. The graph consists of a circular arc representing the electron transfer process and a linear part representing the diffusion process, with the larger radius of the arc defining the more difficult transfer of electrons [63]. The electron transfer ability of the AuNPs/Au electrode is further enhanced than the nanogold deposition previous, leading to the impedance diagram becoming almost straight. Curves c, d, e, and f represent the influence of the modified layer on the electron transfer, which was consistent with the information in the CV diagram.

#### 3.3.3. Standard Curve of Biosensor Measurement

The prepared biosensor can quantitatively detect *Salmonella* mainly attributed to host bacterium that was adsorbed by the phage and formed a complex, which impeded the electron transfer and caused impedance change [64]. The capture time of bacteria affects the concentration of bacteria captured and the detection limit. The biosensor was co-incubated with 2 × 10^6^ CFU/mL bacterial solution every five minutes for 5 to 45 min. Figure 6A showed that the impedance gradually increases within 5~30 min and then a slowed in downward trend after 30 min. According to the results, 30 min was selected as the trapping time for the follow-up study.

The hypothesized reason was attributed to more bacteria being captured by phages when bacterial concentration increases, and when recognition sites were maximally occupied after reaching a certain incubation time, no more bacteria could be immobilized. Meanwhile, if the detection time was too long, the phage would lyse the host bacteria and lead to a decrease in accuracy. Therefore, we found the core functional genes and recombined the receptor binding proteins (RBPs) of phage-based genetic analysis of phage, and BRPs were used as recognition probes to avoid lysis of hosts [65,66]. Hence, our further work of exogenous expression and recombination the RBPs of phage can be used for the detection of pathogenic bacteria [67,68].

The EIS method was used to study the relationship between the concentration of *Salmonella* and the change of resistance (ΔRct). The biosensor was co-incubated with *Salmonella* solutions in the concentration range 20~2.0 × 10^7^ CFU/mL. When data were processed by software CHI660E, the standard curve was plotted with the logarithm of *Salmonella* concentration as the horizontal coordinate and the change in impedance value as the vertical coordinate. The linear regression equation was y = 123.764x + 209.27 and the correlation coefficient was 0.9928 (Figure 6B), respectively. The limit of detection (LOD) of the established method was 13 CFU/mL according to LOD = 3N/S, which has a lower detection line than those published literatures [69,70]. The excellent LOD mainly was due to the synergies of the high specificity of phage, high electron conductivity of AuNPs, and reasonable of biosensor construction.

#### 3.3.4. Specificity of Phage L66-Based Biosensor

The specificity of the prepared biosensor is essential for the application, which was assessed by co-culturing the sensor with different types of host bacteria under the same experimental conditions. The same concentrations (2.5 × 10^6^ CFU/mL, 60 min) of *E. coli*, *S.* Typhimurium, *Listeria*
*monocytogenes*, *S. enteritidis*, *S. indiana*, *S. choleraesuis*, *Vibrio parahemolyticus*, and *S. aureus* were selected for testing. The results of the tests were shown in Figure 7, and all tested strains of *Salmonella* caused significant changes in Rct. In contrast, the non-*Salmonella* strains caused almost no increase in Rct. The same conclusion was obtained from the host spectrum test that phages can specifically identify *Salmonella* host bacteria. The biosensor constructed exhibits more excellence than immunodetection using an electro-optical sensor [71]. The results also validated the rationality of the sensor preparation process.

#### 3.3.5. Stability of Phage L66-Based Biosensor

Stability is a prerequisite for sensor applications, which was assessed by incubating the biosensor with a *Salmonella solution* of 2.5 × 10^6^ CFU/mL with 4 °C or 25 °C and different pH (3~11). As shown in Figure 8A, the ΔRct was essentially unchanged in 15 days, but showed a significant decrease from 15 to 18 days, dropping to about 40% of the original value by 21 days. There was no significant difference between 4 °C and 25 °C, with 4 °C being slightly better than 25 °C. The decrease of ΔRct may be due to a combination of AuNPs shed off the electrode surface, weakening of the amide bond, or inactivation of a small number of phages [72].

The tolerance of the phage L66-based biosensor to different pHs was tested under the same concentration as the bacterial solution of 2.5 × 10^6^ CFU/mL. Before testing, the electrode immerses in PBS buffer at pH 3 to 12 for 1 h, respectively. As shown in Figure 8B, the ΔRct values are almost constant at a pH from 5 to 9. The maximum drop reaches about 30% at the pH of 3 and 12, and decreases about 15% at a pH of 4 and 11. We speculate that the probable reason was the reduced activity of the phage under strong acid and base conditions; at the same time, high hydrogen and hydroxide ions are unfavorable to the electron transfer. Generally, biosensors constructed using phages as recognition probes are more resistant to acids and bases than that constructed with antibodies [73].

#### 3.3.6. Detection of *Salmonella* in Different Substrates

It is necessary to verify the practicality of the phage L66-based biosensor, spiked milk, eggs, and chicken that were processed and used for evaluation of biosensor applications, respectively. *Salmonella* solutions with concentrations of 2.0 × 10^3^, 2.0 × 10^2^, and 20 CFU/mL were added to spiked milk, eggs, and chicken, respectively, and measured three times in parallel to obtain the impedance change values. The concentrations of the bacterial solution were obtained by substituting the standard curve, calculating the recovery range, and finding the relative standard deviation (RSD). The spiked recoveries of *Salmonella* in spiked milk, eggs, and chicken ranged from 94.3% to 109%, from 99.5% to 105.0%, and from 98.5% to 105.5%, respectively, with a relative standard deviation from 3.2% to 6.5%. Specific data were shown in Table 1.

## 4. Conclusions

In this study, a phage strain named L66 was firstly isolated from sewage. The biological properties of phage L66 were analyzed in more detail, and its basic properties were known. This phage has a short latent period and a large burst size, and maintains high reproductive activity over a wide range of pH and thermal stability conditions. Genome-wide association study of phage L66 was performed to predict its potential functions and demonstrated phage L66 is representative of a new genus not previously recognized. Finally, a highly specific biosensor based on the nanogold and phage L66 was constructed, which was applied to the detection of *Salmonella* in spiked milk, eggs, and chicken meat. In the results, it was shown that the established method allows for accurately detecting *Salmonella* of different matrices at a certain level. All in all, this paper presents the preparation, analysis, and application of phage L66, and the strategy provides a theoretical and practical basis for the construction of novel phage-based biosensors. The biosensor based on phage L66 is expected to be a simple, convenient, and low-cost platform for *Salmonella* detection. The platform built has important applications for food safety control and human health [74].

## Data Availability

Data is contained within the article or Appendix A.

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
