# Peer review of "Properties of a Novel Salmonella Phage L66 and Its Application Based on Electrochemical Sensor-Combined AuNPs to Detect Salmonella"

_foods, 2022, doi:10.3390/foods11182836_

Round 1

Reviewer 1 Report

Dear Author

Kindly reduce the plagiarism in the draft to 15. currently it is too high and its similarity with a single source is more than 7%.

what is Meant by LC in source section of supplementary materials?

line two of introduction section says "were the worst" it should be are the worst.

Introduction section first paragraph "In general, treating related diseases requires antibiotics, which were overused and can lead to drug resistance[5, 6]. needs rephrasing. 

Introduction section "In the field of bacteria detection" should be "In the field of bacterial detection"

Introduction last paragraph "and utilized on a gold disc Electrode (GDE) to building a biosensor." should be "and utilized on a gold disc Electrode (GDE) to build a biosensor.

Section 2.2. -layer agar plate method[24, 25]. Purified phage was stored in 20% glycerol with −80℃. it should be "-layer agar plate method[24, 25]. Purified phage was stored in 20% glycerol at −80℃.

section 2.3.2 -"containing LB solution containing 1.5% agar" dont use term containing two times in a single sentence.

section 2.3.4 "The mixture was then co-cultivated at 160 rpm at 37℃ for 45 min. Starting from 0 min, 300 μL of the mixture was taken every 5 min and centrifuged at 8000 rpm for 30 s with 4℃, followed by phage titer was determined and adsorption rate was calculated." kindly rewrite this sentence to make it clear to understand. 

Section 2.3.5. "One-step growth curve can display the incubation period, burst period, and lysis amount of bacteriophage[29]. Dont use term amount rather use some scientific term like copy number.

Same section line 3 "bacterial culture phage" what is meant by bacterial culture phage

section 2.3.6: Phage L66 was co-cultured at various pH and temperatures to evaluate its tolerance against environments. change environments to environmental stresses. 

Section 2.5.2 "In 2.5.1, we processed the GDE and obtained the gold nanoparticles, then the GDE-AuNPs-MPA-Phage L66 composite electrode should be prepared through" instead use was prepared through

same section : Each step was rinsed completely with sterile deionized water.

step can not be rinsed therefore change the sentence structure

Author Response

Response to Reviewer 1 Comments

 Thank you very much for pointing out this issue. The paper has been revised according to your comments. Manuscript has been uploaded to the submission system. The details were as follow:

 Point 1: Kindly reduce the plagiarism in the draft to 15. currently it is too high and its similarity with a single source is more than 7%.

Response 1: The article has been revised in accordance with your suggestions and the requirements of the journal.

 Point 2: what is Meant by LC in source section of supplementary materials?

Response 2: “LC” refers to laboratory collection, a description has been added in Supplementary Materials.

Point 3: line two of introduction section says "were the worst" it should be are the worst.

Response 3: I have revised based on reviewer opinion.

Point 4: Introduction section first paragraph "In general, treating related diseases requires antibiotics, which were overused and can lead to drug resistance[5, 6]. needs rephrasing.

Response 4: I have revised based on reviewer opinion. The revised sentences were as follow: With the overuse of antibiotics, more and more strains of Salmonella resistant bacteria have emerged, posing challenges to clinical medicine and food safety.

 Point 5: Introduction section "In the field of bacteria detection" should be "In the field of bacterial detection"

Response 5: I have revised based on reviewer opinion.

 Point 6: Introduction last paragraph "and utilized on a gold disc Electrode (GDE) to building a biosensor." should be "and utilized on a gold disc Electrode (GDE) to build a biosensor.

Response 6: I have revised based on reviewer opinion.

 Point 7: Section 2.2. -layer agar plate method[24, 25]. Purified phage was stored in 20% glycerol with −80℃. it should be "-layer agar plate method[24, 25]. Purified phage was stored in 20% glycerol at −80℃.

Response 7: I have revised based on reviewer opinion.

Point 8: section 2.3.2 -"containing LB solution containing 1.5% agar" dont use term containing two times in a single sentence.

Response 8: I have revised based on reviewer opinion. The revised sentences were as follow: In brief, initially, the bottom layer was created with LB broth containing 1.5% agar.

Point 9: section 2.3.4 "The mixture was then co-cultivated at 160 rpm at 37℃ for 45 min. Starting from 0 min, 300 μL of the mixture was taken every 5 min and centrifuged at 8000 rpm for 30 s with 4℃, followed by phage titer was determined and adsorption rate was calculated." kindly rewrite this sentence to make it clear to understand.

Response 9: I have revised based on reviewer opinion. The revised sentences were as follow: The mixture was then co-cultivated with 160 rpm at 37℃. Starting from 0 to 45 min, 300 μL of the mixture was taken every 5 min and centrifuged at 8000 rpm for 60 s with 4℃, then phage titer was determined and adsorption rate was calculated.

Point 10: Section 2.3.5. "One-step growth curve can display the incubation period, burst period, and lysis amount of bacteriophage[29]. Dont use term amount rather use some scientific term like copy number.

Response 10: lysis amount has been revised to burst size.

Point 11: Same section line 3 "bacterial culture phage" what is meant by bacterial culture phage

Response 11: bacterial culture phage has been revised to host bacteria.

Point 12: section 2.3.6: Phage L66 was co-cultured at various pH and temperatures to evaluate its tolerance against environments. change environments to environmental stresses.

Response 12: I have revised based on reviewer opinion.

Point 13: Section 2.5.2 "In 2.5.1, we processed the GDE and obtained the gold nanoparticles, then the GDE-AuNPs-MPA-Phage L66 composite electrode should be prepared through" instead use was prepared through

Response 13: I have revised based on reviewer opinion.

 Point 14: same section : Each step was rinsed completely with sterile deionized water.

step can not be rinsed therefore change the sentence structure

Response 14: Sentences have been rewritten. The revised sentences were as follow: The electrodes were rinsed completely with sterile deionized water after measuring different concentrations of phage.

Thank you again for all your hard work and send my best wishes.

Reviewer 2 Report

The manuscript is an article about the detection of Salmonella by using a highly specific phage-based biosensor. The topic is consistent with the scope of the journal and the manuscript is generally well structured. The research is detailed, material and methods are well performed and the conclusions are of interest. The study is related to previous research in this area and to recent literature. However, the article need changes in typos and spelling and a revision of the introduction section. Specific suggestions are reported in the attached file.

Author Response

Response to Reviewer 2 Comments

Thank you very much for pointing out this issue. The paper has been revised according to your comments. Manuscript has been uploaded to the submission system. The details were as follow:

Point 1: Line 35: Please replace Salmonella Enterica with Salmonella enterica.

Response 1: I have revised based on reviewer opinion.

Point 2: Line 34-39: I suggest you to amplify the introduction part, explaining better the problem related to Salmonella in food and its multidrug resistance. You can check and refer to “The European Union Summary Report on Antimicrobial Resistance in zoonotic and indicator bacteria from humans, animals and food in 2019–2020, EFSA”. According to this report, Campylobacteriosis is the most reported zoonosis in the EU, followed by Salmonellosis. Please cite: Pepe, T., De Dominicis, R., Esposito, G., Ventrone, I., Fratamico, P. M., & Cortesi, M. L. (2009). Detection of Campylobacter from poultry carcass skin samples at slaughter in Southern Italy. Journal of food protection, 72(8), 1718-1721.

Response 2: I have revised some sentences based on reviewer opinion and replaced references.

Point 3: Line 65: Please replace the sentence with “Luria−Bertani (LB) Broth and Agar were obtained..”

Response 3: I have revised based on reviewer opinion.

Point 4: Line 69: Please replace the sentence “was obtained” with “were obtained”.

Response 4: I have revised based on reviewer opinion.

Point 5: Line 77: Please replace “Salmonella Typhimurium” with ““Salmonella Typhimurium”.

Response 5: I have revised based on reviewer opinion.

Point 6: Line 78: Please specify what is LA solution.

Response 6:  I have revised that LB broth(containing 0.7% Agar).

Point 7: Line 79: What do you mean for solution? Clarify the concept.

Response 7: solution means broth, i have revised.

Point 8: Line 83: Please replace “with −80℃” with “at −80℃”.

Response 8: I have revised based on reviewer opinion.

Point 9: Line 94: Please replace the sentence with “the bottom layer was created with LB broth containing 1.5% agar…”

Response 9: I have revised based on reviewer opinion.

Point 10: Line 96: Please add “broth” to “LB”

Response 10: I have revised based on reviewer opinion.

Point 11:  Line 110: Please replace “followed by” with “then”

Response 11: I have revised based on reviewer opinion.

Point 12: Line 129: Please capitalize the first letter.

Response 12: I have revised based on reviewer opinion.

Point 13: Line 189: Please replace “Materials” with “food matrices or food samples”.

Response 13: I have revised the “Materials” with “food samples”.

Point 14: Line 277: Please replace the comma with the spot.

Response 14: I have revised based on reviewer opinion.

Point 15: Line 467: Please delete the “isolated” repetition.

Response 15: I have revised based on reviewer opinion.

Point 16: Line 467-480: Please add some references to the importance of this research in the food safety field.

Response 16: I have added references on the food safety field in the conclusion section.

Thank you again for all your hard work and send my best wishes.
